# The Effects of *Crataegus pinnatifida* and *Wolfiporia extensa* Combination on Diet-Induced Obesity and Gut Microbiota

**DOI:** 10.3390/foods13111633

**Published:** 2024-05-24

**Authors:** Jingjing Yuan, Yueyun Hu, Dongmei Yang, An Zhou, Shengyong Luo, Na Xu, Jiaxing Dong, Qing He, Chenxu Zhang, Xinyu Zhang, Zhangxin Ji, Qinglin Li, Jun Chu

**Affiliations:** 1Key Laboratory of Xin’an Medicine, Ministry of Education, Anhui University of Chinese Medicine, Hefei 230012, China; yjgogo123@ahtcm.edu.cn (J.Y.); hyy1230426@163.com (Y.H.); 13215533945@163.com (D.Y.); anzhou@ahtcm.edu.cn (A.Z.); dongjiaxing2024@163.com (J.D.); heqing@stu.ahtcm.edn.cn (Q.H.); zcx18715092956@163.com (C.Z.); xinyu@stu.ahtcm.edu.cn (X.Z.); jizhangxin@163.com (Z.J.); 2Research and Technology Center, Anhui University of Chinese Medicine, Hefei 230012, China; 3Functional Activity and Resource Utilization on Edible and Medicinal Fungi Joint Laboratory of Anhui Province, Anhui University of Chinese Medicine, Hefei 230012, China; 4Affiliated Hospital of Yangzhou University, Yangzhou 225012, China; 5Anhui Academy of Medical Sciences, Hefei 230061, China; lsy770728@126.com; 6State Key Laboratory of Tea Plant Biology and Utilization, School of Tea and Food Science and International Joint Laboratory on Tea Chemistry and Health Effects of Ministry of Education, Anhui Agricultural University, Hefei 230036, China; naxu2014@ahau.edu.cn; 7Institute of Surgery, Anhui Academy of Chinese Medicine, Anhui University of Chinese Medicine, Hefei 230012, China

**Keywords:** *Crataegus pinnatifida*, *Wolfiporia extensa*, obesity, high-fat diet, gut microbiota

## Abstract

Obesity is a multifactorial chronic metabolic disease with multiple complications. *Crataegus pinnatifida* (CP) and *Wolfiporia extensa* (WE) are traditional functional foods with improving metabolic health properties. This study demonstrated the effect of CP and WE combination on ameliorating obesity induced by a high-fat diet (HFD). Moreover, the CP-WE food pair ameliorated HFD-induced metabolic disorders, including glucose intolerance, insulin resistance, hyperlipidemia, and hepatic steatosis. 16S rRNA gene amplicon sequencing and analysis revealed that CP combined with WE reshaped the composition of gut microbiota in HFD-fed mice. Furthermore, correlation analysis revealed a substantial association between the obesity-related parameters and the shifts in predominant bacterial genera influenced by the food pair intervention. In conclusion, this study demonstrated that the CP-WE food pair ameliorated HFD-induced obesity and reshaped gut microbiota composition, providing a promising approach to combat obesity through specific food combinations.

## 1. Introduction

Obesity is a modern metabolic disorder characterized by excessive lipid accumulation. It is associated with a range of complications, including type 2 diabetes, dyslipidemia, and hepatic steatosis [1]. Given its increasing prevalence, there is an urgent need to explore safe and effective anti-obesity interventions.

Gut microbiota plays a crucial role in maintaining metabolic health [2,3]. Mounting evidence suggests a correlation between the development of obesity and gut microbial dysbiosis [4]. Various determinants, encompassing age, lifestyle, dietary habits, and host genetics, collectively influence the composition of gut microbiota. Among these factors, diet appears to be one of the most modifiable determinants shaping gut microbiota composition [5]. In 2000, Daubioul et al. discovered that soluble fiber-induced changes in the gut microbiota decreased adipose mass and hepatic steatosis in obese *fa/fa* Zucker rats [6]. Subsequently, it was reported that the high-fat diet (HFD) promoted the growth of microbial communities associated with obesity [7]. Conversely, diets rich in prebiotics prevented the development of obesity [4,8]. Therefore, dietary interventions targeting gut microbiota have emerged as novel strategies for the management of obesity.

Recently, the concept of “food as medicine” has gained recognition [9]. However, attempts to ameliorate metabolic diseases with a single food or nutrient intervention have been met with limited success. Edible herbs, also known as “food-medicine dual-use” foods, possess both dietary properties and pharmacological functions. According to traditional Chinese medicine (TCM) principles, specific combinations of medicinal foods can have synergistic effects [10,11]. *Crataegus pinnatifida* (CP), known as Shan Zha or hawthorn, is a popular functional food in China. It has a tart-sweet taste and is abundant in phenols, flavonoids, terpenoids, fiber, and organic acids [12,13,14]. It has been documented that these chemical constituents exhibit a wide range of pharmacological effects, including anticancer properties, antioxidant activity, improvement of glycolipid metabolism, and facilitation of digestive processes [12,13,14,15,16]. *Wolfiporia extensa* (WE), or Fu Ling, is an edible fungus with a long consumption history. It contains multiple bioactive components, including polysaccharides, triterpenoids, and steroids [17,18,19,20,21]. These components demonstrate pharmacological effects such as diuretic properties, immunomodulatory activity, anti-tumor activity, and hepatoprotection [22,23,24,25].

CP and WE have been widely used in TCM since two thousand years ago due to their medicinal properties. CP is traditionally used to promote digestion, improve circulation, and regulate lipid metabolism [12]. WE is known for its sedative, diuretic, and spleen-boosting properties [26]. Recent studies have demonstrated that CP extracts, including polysaccharides, corosolic acid, hyperoside, and maslinic acid, protect against HFD-induced obesity and nonalcoholic fatty liver disease [27,28,29,30]. In addition, WE polysaccharides and triterpenoids have also been reported to reduce lipid accumulation and ameliorate hepatic steatosis in obese mice [18,31,32]. However, the synergistic effect of the combination of CP and WE on ameliorating obesity and its underlying mechanism requires more investigation.

In this study, we investigated the potential synergistic effect of CP and WE on HFD-induced obesity. Moreover, the effects of the food pair on HFD-induced metabolic disorders were evaluated. Subsequently, 16S rRNA gene amplicon sequencing and bioinformatics analysis were performed to explore the effects of the food pair on the gut microbiome composition. Finally, we analyzed the correlation between the obesity-related parameters and the predominant bacterial genera affected by the food pair intervention.

## 2. Materials and Methods

### 2.1. Materials

CP and WE were procured from Beijing Tongrentang Co., Ltd. (Beijing, China) and verified by Professor Qinglin Li (Anhui University of Chinese Medicine, Hefei, China). The organic acid content of CP is >4%, and the ethanol extract of WE is >2.5%, which conforms to the provisions of the Chinese Pharmacopoeia. The specimens (voucher NO. 20210730 and 20210731) have been deposited in the Key Laboratory of Xin’an Medicine, Ministry of Education, Anhui University of Chinese Medicine (Hefei, China).

### 2.2. Animals and Experimental Design

Male C57BL/6 mice were purchased from Vital River Laboratory Animal Technology Co., Ltd. (Beijing, China). At 5 weeks old, the mice were randomly divided into five groups: low-fat diet (LFD, n = 8), HFD (45% fat, n = 8), HFD+CP (HFD supplemented with 2.5 g/kg/d CP, n = 8), HFD+WE (HFD supplemented with g/kg/d WE, n = 8), and HFD+CPWE (HFD supplemented with 2.5 g/kg/d CP and 5 g/kg/d WE, n = 8), as shown in Figure 1. The compositions of diets are shown in Table 1. The body weight and food intake of mice were monitored weekly. After a 12-week treatment, the blood, intestinal contents, liver, and adipose tissue samples were collected. All murine experiments were approved by the Anhui University of Chinese Medicine Animal Care and Use Committee (AHUCM-mouse-2021120).

### 2.3. GTT and ITT

GTT and ITT were conducted as previously described [35]. In brief, for the glucose tolerance tests, mice received intraperitoneal injections of glucose (1 g/kg body weight) after overnight fasting. Glucose levels were assessed using an ACCU-CHEK Performa glucometer (Roche, Mannheim, Germany) at 0, 15, 30, 45, 60, 90, and 120 min post-injection. For the insulin tolerance tests, mice were fasted for six hours, followed by intraperitoneal administration of insulin (1.5 UI/kg bodyweight). Subsequent to the injection, blood glucose levels were monitored at 0, 15, 30, 45, 60, 90, and 120 min post-injection.

### 2.4. Biochemical Analysis

The fresh blood samples were incubated at 4 °C for four hours to achieve stability, followed by centrifugation at 5000× *g* for 15 min. Levels of serum total cholesterol (TC), high-density lipoprotein cholesterol (HDL-C), low-density lipoprotein cholesterol (LDL-C), triglyceride (TG), alanine aminotransferase (ALT), and aspartate aminotransferase (AST) were detected using the commercial kits (Nanjing Jiancheng, Nanjing, Jiangsu, China), according to the instructions. 

### 2.5. Histological Analysis

The liver samples were fixed in a 4% paraformaldehyde solution for 24 h, embedded in paraffin, and then sectioned into 5 μm thick slices. Following this, all slides underwent deparaffinization, rehydration, and standard hematoxylin and eosin (H&E) staining procedures. Slides were digitalized using a panoramic digital scanner (3D Histech, Budapest, Hungary). The assessment of the images was conducted using CaseViewer 2.0 software (3D Histech, Budapest, Hungary).

### 2.6. 16S rRNA Gene Sequencing and Analyses

The cecal contents were promptly collected and immediately frozen. Microbial DNA was extracted from frozen samples with a commercial kit (MP Biomedicals, Santa Ana, CA, USA). The specific primers (Forward: 5′-CCTACGGGNGGCWGCAG-3′, Reverse: 5′-GGACTACHVGGGTWTCTAA-3′) were used to amplify the bacterial gene. The library was constructed and paired-end sequenced by the Illumina NovaSeq platform (Illumina, San Diego, CA, USA).

Microbiome bioinformatics analysis was conducted using QIIME2, according to the official tutorials. Briefly, sequence data underwent rigorous quality filtering, denoising, merging, and elimination of chimeric sequences using the DADA2 plugin. Taxonomic assignments of amplicon sequence variants (ASVs) were conducted based on the Greengenes database. In addition, the Vsearch (v2.13.4) was used for the operational taxonomic unit (OTU) clustering. Alpha diversity was analyzed with Chao1, Observed_species, Shannon, and Simpson indices. Beta diversity was estimated based on the Bray–Curtis dissimilarity. Biomarkers were identified through LEfSe analysis. The association between obesity-related parameters and the relative abundance of dominant genera was assessed using Spearman’s rank correlation analysis.

### 2.7. Statistic Analysis

The data were presented as mean ± SE, and statistical analysis was performed using SPSS 22 (SPSS, Chicago, IL, USA). Tukey’s test was applied to assess differences among the groups. For beta diversity, statistical differences between microbial communities were determined by ANOSIM and PERMANOVA with the Bonferroni test. Figures were plotted using Origin 2022 (QriginLab, Northampton, MA, USA) and Biorender (Biorender, Toronto, ON, Canada). All statistical tests with *p* < 0.05 were considered statistically significant.

## 3. Results

### 3.1. Effects of CP and WE on HFD-Induced Obesity

As shown in Figure 1, the effects of CP and WE on diet-induced obesity were evaluated in C57BL/6 mice. After a 12-week treatment, the bodyweight gain of HFD-fed mice was found to be significantly higher than that of LFD-fed mice (Figure 2a,b). The WE-treated mice had an even higher bodyweight gain than the HFD group (Figure 2a,b). Meanwhile, CP treatment could significantly reduce the bodyweight gain in HFD-fed mice (Figure 2a,b). Notably, the combination of CP and WE (CPWE) had a more significant effect on ameliorating diet-induced bodyweight gain (Figure 2a,b). In addition, supplementation of CP and WE separately or in combination could not significantly affect the energy intake of mice (Figure 2c).

The development of obesity is accompanied by lipid accumulation in adipose tissue [33,36]. We then assessed adipose tissue weight and found that supplementation of CP or WE individually could not significantly reduce the weight of epididymal adipose tissue (EAT) (Figure 2d), subcutaneous adipose tissue (SAT) (Figure 2e), or brown adipose tissue (BAT) (Figure 2f). However, the supplementation of CP and WE together could significantly suppress lipid deposition in adipose tissues (Figure 2d–f).

These findings suggest that the combined administration of CP and WE exerts a synergistic effect in attenuating diet-induced bodyweight gain.

### 3.2. Effects of CPWE on Glucose Intolerance and Hyperlipidemia in HFD-Fed Mice

Diet-induced obesity is typically accompanied by impaired glucose tolerance and insulin resistance [37]. Hence, we examined the effects of CPWE on obesity-induced glucose metabolic disorders. Consistent with lower body weight, the CPWE-treated mice had lower fasting glucose levels (Figure 3a). Moreover, GTT and ITT results showed that CPWE effectively reversed HFD-induced glucose intolerance (Figure 3b,c) and insulin resistance (Figure 3d,e).

Hyperlipidemia is a common complication of diet-induced obesity [38]. In this study, HFD-fed mice showed significantly higher serum TG (Figure 3f), TC (Figure 3g), LDL-C (Figure 3h), and HDL-C (Figure 3i) levels compared to those on an LFD, while dietary CPWE dramatically depressed these serum lipid levels in HFD-fed mice (Figure 3f–i).

These findings demonstrate that the CP-WE food pair ameliorates HFD-induced glucose intolerance, insulin resistance, and hyperlipidemia.

### 3.3. Effects of CPWE on Liver Lipid Deposition in HFD-Fed Mice

In obesity, excess fat accumulation causes adipose tissue dysfunction, leading to the ectopic accumulation of lipids in non-adipose tissues [37]. The ectopic overaccumulation of lipids in the liver is correlated with hepatic steatosis and abnormal hepatic function. Figure 4a shows that the liver of LFD-fed mice displayed a healthy red hue, while the liver of HFD-fed mice appeared pale and contained fat particles. Notably, dietary supplementation of CPWE could effectively ameliorate HFD-induced fatty liver. Moreover, consistent with the lower liver weight (Figure 4b), H&E histological staining showed that CPWE sufficiently reversed hepatic lipid deposition in HFD-fed mice (Figure 4c). In addition, CPWE also decreased serum ALT (Figure 4d) and AST (Figure 4e) levels. These results indicate that CPWE ameliorates HFD-induced hepatic steatosis.

### 3.4. Effects of CPWE on Gut Microbiota Composition in HFD-Fed Mice

The development of obesity is associated with gut microbial dysbiosis [4]. Hence, we explored the effects of CPWE on gut microbiota composition. The alpha diversity of gut microbiota was analyzed with the Chao1, Obeserved_species, Simpson, and Shannon indices. There was no significant difference in gut microbiota richness and diversity between the HFD and HFD+CPWE groups (Figure 5a–d). In addition, principal coordinate analysis (PCoA) and nonmetric multidimensional scaling (NMDS) score plots were used to assess the beta diversity of gut microbiota. Both PCoA and NMDS analyses showed a considerable separation among the groups (Figure 5e,f). Statistical analysis of beta diversity showed a significant difference between the HFD and HFD+CPWE groups (PERMANOVA: F = 17.250415, *p* = 0.002, and ANOSIM: R = 1.000000, *p* = 0.002).

At the phylum level, HFD significantly increased the relative abundance of *Firmicutes* in mice (Figure 6a,b). Compared to the HFD group, the CPWE-treated mice had a mild increase in the abundance of *Firmicutes* (Figure 6a,b). Additionally, no significant difference was observed in the relative abundance of the *Firmicutes*/*Bacteroidetes* ratio (Figure 6c) among the three groups. At the genus level, the decreased abundance of *Allobaculum* in HFD-fed mice was significantly reversed by CPWE supplementation (Figure 6d,e). In addition, CPWE also increased the relative abundance of *Anaerostipes*, *Oscillospira*, *Prevotella*, and *Dehalobacterium* compared to the HFD-fed mice (Figure 6f–i). Meanwhile, the CPWE-treated mice had a significantly lower abundance of *Parabacteroides*, *Blautia*, and *Streptococcus* than the HFD group (Figure 6j–l). 

These results reveal that CPWE reshapes the gut microbiome composition in HFD-fed mice.

### 3.5. Analysis of Specific Dominant Bacterial Phenotypes

To further investigate the gut microbiota composition and the specific dominant bacterial phenotypes of each group, the numbers of ASVs that are specific or shared among the three groups were represented using Venn analysis (Figure 7a). Additionally, the taxonomic analysis showed that the specific dominant phylum of the HFD+CPWE group was *Firmicutes*, followed by *Bacteroidetes* (Figure 7b). At the genus level, the relative abundance of *Allobaculum*, *Oscillospira*, and *Anaerostipes* together accounted for more than 70% of the specific bacteria in the HFD+CPWE group (Figure 7c). Moreover, the heatmap of the 50 most abundant gut bacterial genera indicated that CPWE supplementation decreased the abundance of *Parabacteroides*, *Blautia*, *Streptococcus*, *Anaerotruncus*, and *Christensenella* and increased the abundance of *Allobaculum*, *Anaerostipes*, *Oscillospira*, *Prevotella*, *Dehalobacterium*, *Turicibacter*, *Cetobacterium*, and *Bacteroides* in HFD-fed mice (Figure 7d–g).

To identify the specific enriched bacterial genera, the LEfSe analysis was performed with LDA scores greater than 3.0 (Figure 8a). A total of 25 enriched bacteria at the genus level were screened out from the LFD, HFD, and HFD+CPWE groups. Moreover, six of them were the most abundant genera (LDA score > 4.0). *Akkermansia*, *Adlercreutzia*, and *Coprococcus* were the dominant genera in the LFD group. The most significantly enriched genus in HFD-fed mice was *Parabacteroides*. *Allobaculum* and *Anaerostipes* were the dominant genera of the HFD+CPWE group. Consistently, the taxon significantly enriched in abundance was also shown in the taxonomic cladogram from phylum to genus levels (Figure 8b).

### 3.6. Correlation between the Obesity-Related Parameters and the Predominant Bacterial Genera

To evaluate the association between the bacterial genera and obesity-related parameters, a Spearman’s rank correlation analysis was performed. In total, 20 dominant genera had a significant correlation with obesity-associated parameters, including body weight, adipose tissue mass, liver weight, and serum levels of glucose, TG, TC, HDL-C, LDL-C, ALT, and AST (Figure 9). The relative abundance of *Allobaculum*, *Oscillospira*, *Dehalobacterium*, *Turicibacter*, and *Prevotella* that increased by CPWE supplementation were negatively correlated with obesity-related parameters. Conversely, the decreased genera, including *Parabacteroides*, *Blautia*, *Streptococcus*, *Anaerotruncus*, and *Christensenella*, had a positive correlation with obesity-related parameters (Figure 9). These results suggest that the dominant bacterial genera affected by CPWE supplementation are closely associated with HFD-induced obesity.

## 4. Discussion

Obesity is a chronic metabolic disease characterized by excess lipid accumulation [37]. Adipose tissue is considered a major storage site of TG. In the development of HFD-induced obesity, the overaccumulation of fat triggers ectopic lipid accumulation. Excess lipid accumulation in circulation and non-adipose tissues can cause lipotoxicity and metabolic dysfunction, including hyperlipidemia, hepatic steatosis, and insulin resistance [39]. In this study, CP and WE individually could not significantly reduce the adipose tissue weight. However, the combination of CP and WE reversed HFD-induced overaccumulation of fat in adipose tissues (Figure 2d–f). Moreover, dietary supplementation with CPWE significantly reduced serum lipid levels and hepatic lipid deposition, as well as improved glucose tolerance and insulin resistance in HFD-fed mice (Figure 3 and Figure 4). These findings demonstrate that the CP-WE food pair not only prevents HFD-induced obesity but also ameliorates the related metabolic disorders.

TCM adopts a holistic view of healthcare, emphasizing the equilibrium and coordination of systems. This comprehensive perspective has proven particularly advantageous in effectively managing chronic metabolic disorders, including obesity. One advantage of TCM treatments is their synergistic use of herbs, which possess multiple bioactive compounds. The synergistic interplay of these bioactive constituents can amplify therapeutic effects in clinical practice. It has been reported that CP and WE contain multiple constituents with lipid-lowering activity. For instance, a flavonoid glycoside component of CP, hyperoside, has been reported to ameliorate diet-induced obesity by facilitating adipose tissue browning and lipophagy [27]. Maslinic acid, a pentacyclic triterpene acid in CP, has been shown to protect against nonalcoholic fatty liver through the Sirt1/AMPK pathway in mice [30]. Corosolic acid, another pentacyclic triterpenoid in CP, can ameliorate nonalcoholic steatohepatitis in HFD-fed mice [28]. In addition, WE polysaccharides and triterpenoids have also been reported to reduce lipid accumulation and ameliorate hepatic steatosis in obese mice [18,22,32]. Given the complex synergistic interactions of multiple bioactive constituents, the detailed mechanisms underlying the synergistic anti-obesity effect of the food pair merit further investigation.

Although it has been acknowledged that gut microbiota dysbiosis is associated with the development of obesity, the exact mechanism of how gut microbiota affects body weight remains elusive. In 2006, Ley et al. proposed that the elevated *Firmicutes*/*Bacteroidetes* ratio could be a putative characteristic of obesity [40]. However, this hypothesis was soon challenged by Duncan et al. as they did not detect a significant difference in the *Firmicutes*/*Bacteroidetes* ratio between lean and obese individuals [41]. In this study, no significant difference in the *Firmicutes*/*Bacteroidetes* ratio was observed among groups (Figure 6j). Hence, the *Firmicutes*/*Bacteroidetes* ratio does not seem to be a reliable indicator in the management of obesity. Consistent with previous reports [42,43,44,45], we found that the abundance of *Allobaculum* was significantly decreased in HFD-fed mice. However, the abundance of *Parabacteroides*, *Blautia*, *Streptococcus*, and *Anaerotruncus*, which were positively correlated with obesity, was significantly increased. Interestingly, CPWE reversed these changes in the abundance of obesity-associated bacterial genera (Figure 6, Figure 7, Figure 8 and Figure 9). These findings suggest that the CP-WE food pair effectively ameliorates HFD-induced gut microbial dysbiosis.

It is intriguing to note that the dynamic balance of gut microbiota is essential for maintaining metabolic health [5], a holistic perspective that dovetails with the emphasis on systemic balance in TCM. Numerous factors should be considered when exploring the role of gut microbiota in the synergistic anti-obesity effects of CP and WE. The bioactive constituents in CP and WE have beneficial effects on gut microbiota [46,47,48,49,50]. For instance, the procyanidin extract of CP down-regulated the abundance of *Parabacteroides* and up-regulated *Turicibacter* during in vitro digestion and fermentation [49]. Zou et al. reported that WE and its polysaccharide fractions remarkably decreased the abundance of *Parabacteroides* and *Blautia* while increasing the abundance of *Allobaculum* [50]. Moreover, WE oligosaccharides increased the abundance of *Turicibacter* in colitis mice [26]. Thus, we speculated that the synergistic anti-obesity effect of CP and WE is, at least partially, associated with the gut microbiota. However, given the complex interactions of multiple bioactive constituents and gut microbiota, more evidence is required to unravel the precise mechanisms underlying these interactions.

## 5. Conclusions

In conclusion, this study demonstrates that the CP and WE food pair synergistically prevents HFD-induced obesity. Moreover, the food pair ameliorates metabolic disorders and reshapes the gut microbiota in HFD-fed mice. Correlation analysis reveals a significant association between the obesity-associated parameters and the dominant bacterial genera affected by CPWE supplementation. The present study provides a promising approach to combat obesity through specific food combinations. 

## Figures and Tables

**Figure 1 foods-13-01633-f001:**
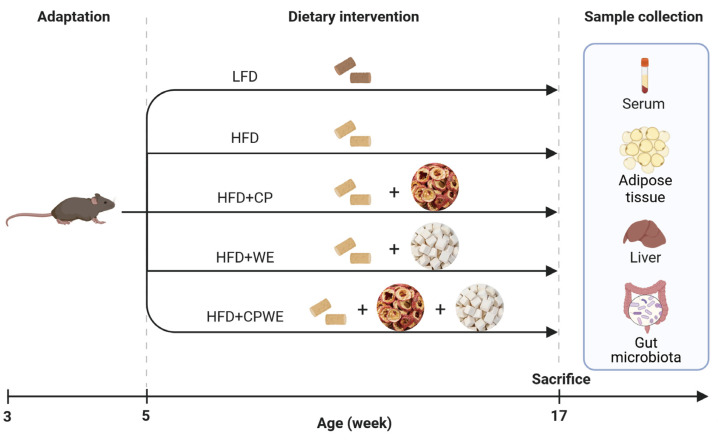
The schematic of animal experimental design. LFD, low-fat diet; HFD, high-fat diet; CP, *Crataegus pinnatifida*; WE, *Wolfiporia extensa*; CPWE, *Crataegus pinnatifida* and *Wolfiporia extensa*.

**Figure 2 foods-13-01633-f002:**
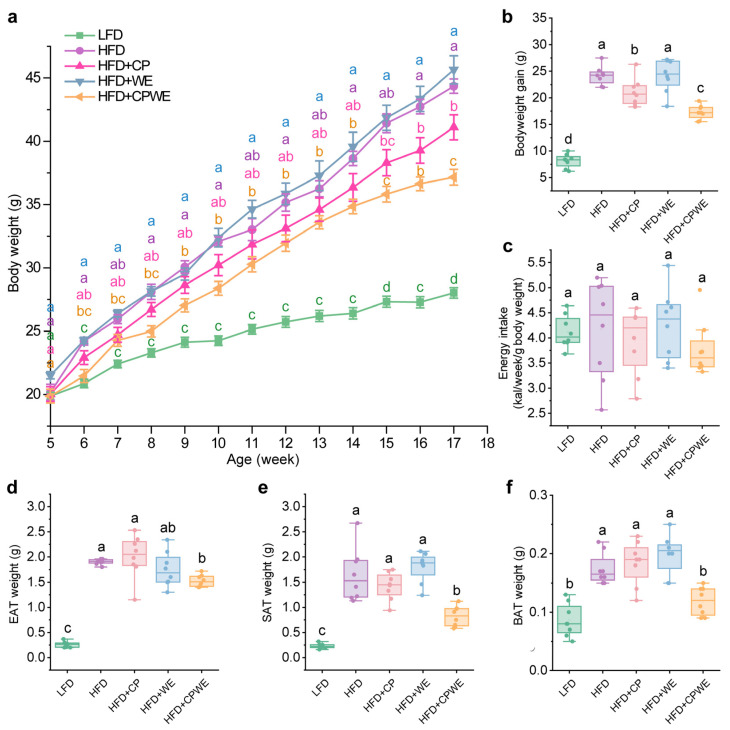
(**a**–**c**) Body weight (**a**), bodyweight gain (**b**), and energy intake (**c**) of mice with the indicated treatments (n = 8). (**d**–**f**) Weight of EAT (**d**), SAT (**e**), and BAT (**f**) from mice with the indicated treatments (n = 8). EAT, epididymal adipose tissue; SAT, subcutaneous adipose tissue; BAT, brown adipose tissue. Statistical differences among groups were indicated by the different lowercase letters (Tukey’s test, *p* < 0.05).

**Figure 3 foods-13-01633-f003:**
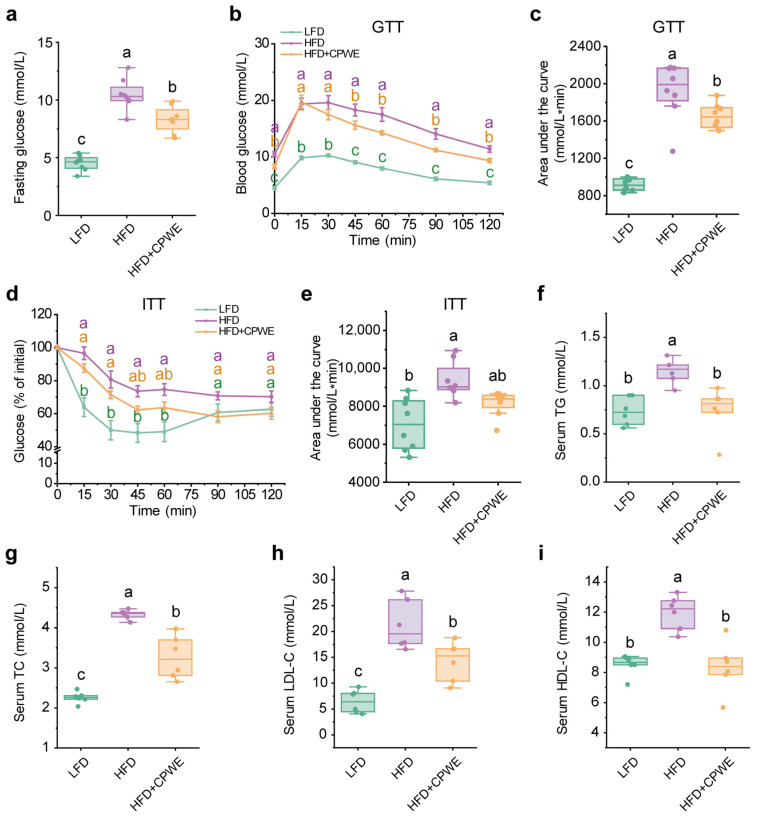
(**a**–**e**) Fasting glucose (**a**), GTT (**b**), AUC of GTT (**c**), ITT (**d**), and AUC of ITT (**e**) of mice with the indicated treatments (n = 8). (**f**–**i**) Serum levels of TG (**f**), TC (**g**), LDL-C (**h**), and HDL-C (**i**) in mice (n = 6). AUC, area under the curve; GTT, glucose tolerance test; ITT, insulin tolerance test; TG, triglyceride; TC, total cholesterol; LDL-C, low-density lipoprotein cholesterol; HDL-C, high-density lipoprotein cholesterol. Statistical differences among groups were indicated by the different lowercase letters (Tukey’s test, *p* < 0.05).

**Figure 4 foods-13-01633-f004:**
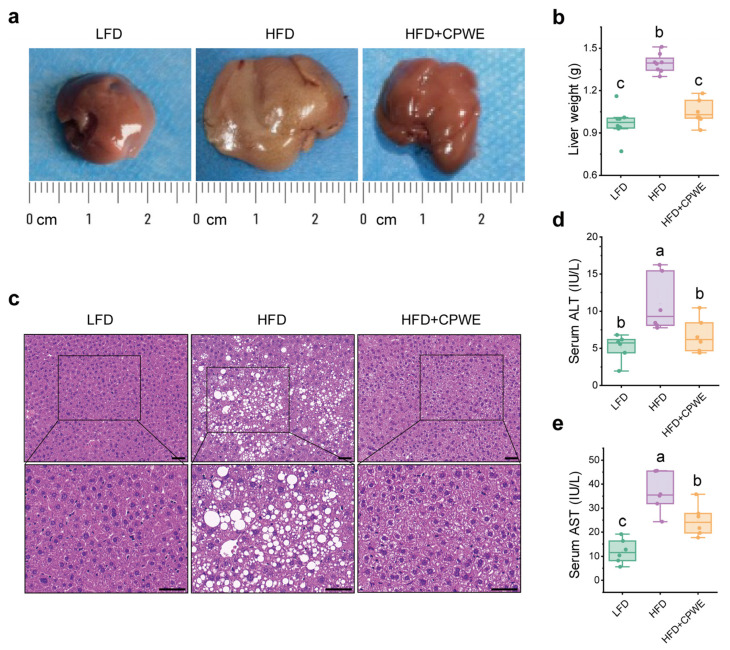
(**a**–**c**) Representative photographs (**a**), weight (**b**), and H&E staining (**c**) of liver from mice with the indicated treatments (n = 8). (**d**,**e**) Serum levels of ALT (**d**) and AST (**e**) in mice (n = 6). Scale bar, 50 μm. H&E, hematoxylin and eosin; ALT, alanine aminotransferase; AST, aspartate aminotransferase. Statistical differences among groups were indicated by the different lowercase letters (Tukey’s test, *p* < 0.05).

**Figure 5 foods-13-01633-f005:**
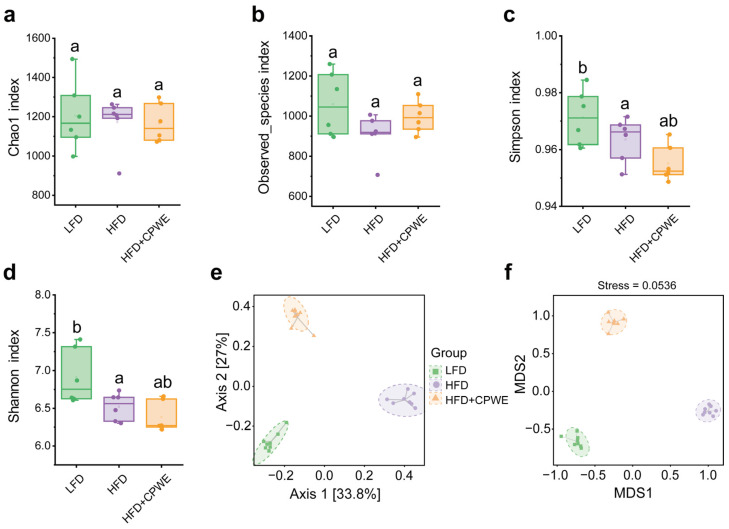
(**a**–**d**) The alpha diversity indices of Chao1 (**a**), Obeserved_species (**b**), Simpson (**c**), and Shannon (**d**) (n = 6). (**e**,**f**) The PCoA (**e**) and NMDS (**f**) analyses (n = 8). PCoA, principal coordinate analysis; NMDS, nonmetric multidimensional scaling. Statistical differences among groups were indicated by the different lowercase letters (Tukey’s test, *p* < 0.05).

**Figure 6 foods-13-01633-f006:**
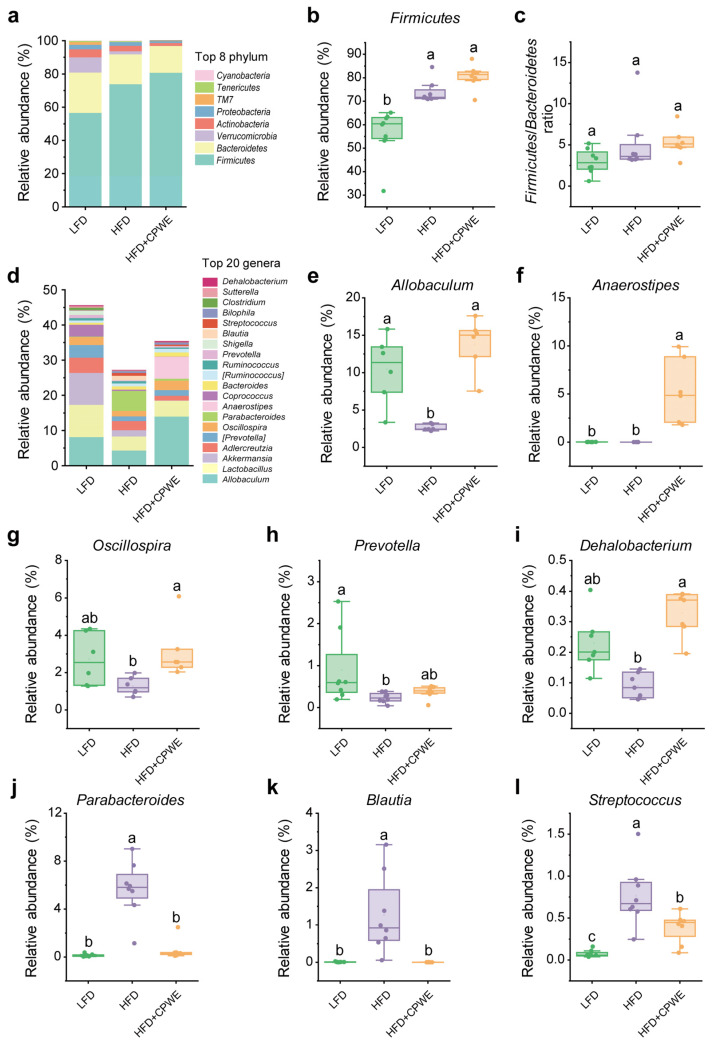
(**a**) The relative abundance of the top eight phylum (n = 8). (**b**,**c**) The relative abundance of *Firmicutes* (**b**) and the *Firmicutes*/*Bacteroidetes* ratio (**c**) in mice with the indicated treatments (n = 8). (**d**) The relative abundance of the top 20 genera (n = 8). (**e**–**l**) The relative abundance of *Allobaculum* (**e**), *Anaerostipes* (**f**), *Oscillospira* (**g**), *Prevotella* (**h**), *Dehalobacterium* (**i**), *Parabacteroides* (**j**), *Blautia* (**k**), and *Streptococcus* (**l**) in mice (n = 8). Statistical differences among groups were indicated by the different lowercase letters (Tukey’s test, *p* < 0.05).

**Figure 7 foods-13-01633-f007:**
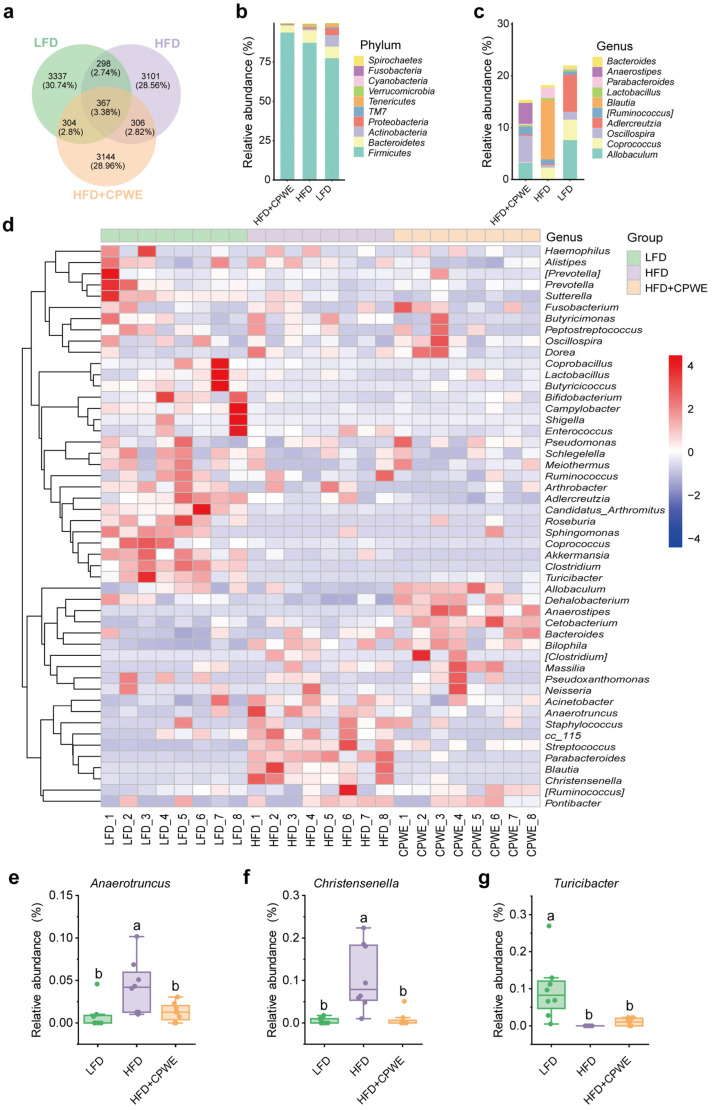
(**a**) Venn analysis of the gut microbiota composition (n = 8). (**b**,**c**) Relative abundance of the specific dominant phylum (**b**) and genera (**c**) in the indicated groups (n = 8). (**d**) Heatmap of the top 50 genera in mice (n = 8). Colors range from red (relative abundance increased) to blue (relative abundance reduced). (**e**–**g**) The relative abundance of *Anaerotruncus* (**e**), *Christensenella* (**f**), and *Turicibacter* (**g**) in mice (n = 8). Statistical differences among groups were indicated by the different lowercase letters (Tukey’s test, *p* < 0.05).

**Figure 8 foods-13-01633-f008:**
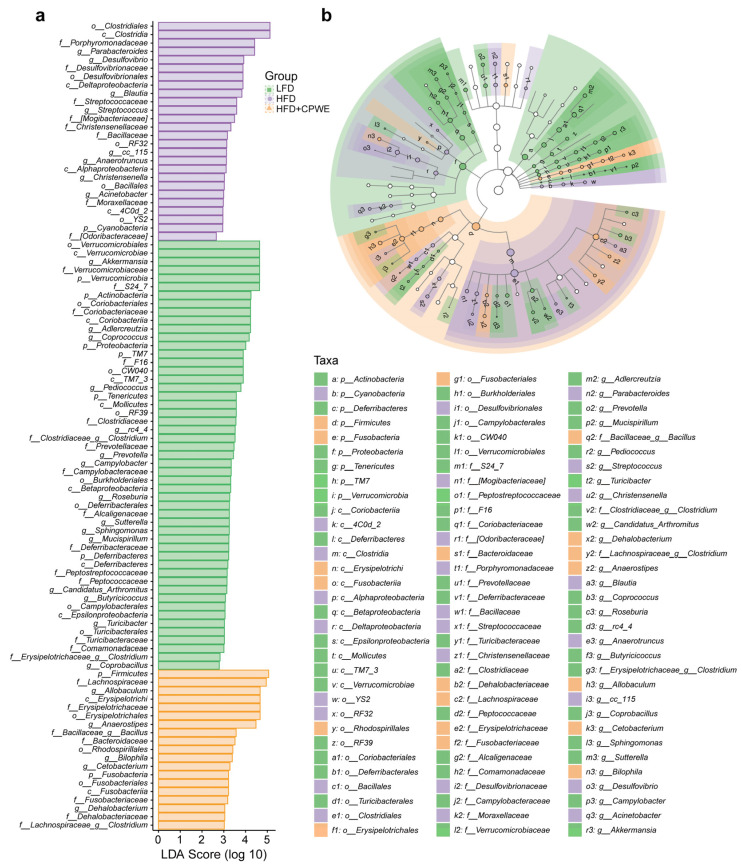
(**a**,**b**) The LDA scores (**a**) and cladogram (**b**) of enriched gut microbiota from the phylum to genus levels. LDA score > 3.0. LDA, linear discriminant analysis.

**Figure 9 foods-13-01633-f009:**
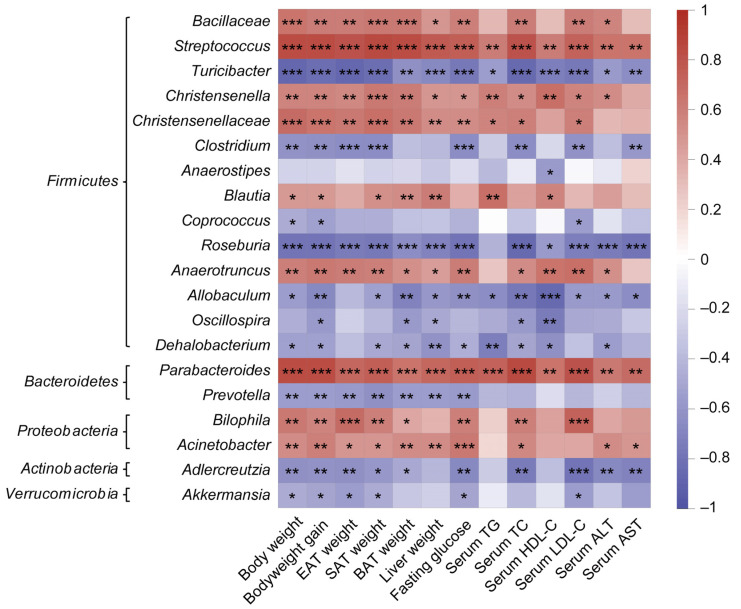
Heatmap of correlation analysis. The genera are grouped by phyla *Firmicutes*, *Bacteroidetes*, *Proteobacteria*, *Actinobacteria*, and *Verrucomicrobia*. * *p* < 0.05, ** *p* < 0.01, *** *p* < 0.001.

**Table 1 foods-13-01633-t001:** The compositions of diets.

Ingredient (g)	LFD	HFD	HFD+CP	HFD+WE	HFD+CPWE
Casein	200	200	200	200	200
Maltodextrin 10	35	100	100	100	100
Lard	20	177.5	177.5	177.5	177.5
Calcium carbonate	5.5	5.5	5.5	5.5	5.5
Cellulose	50	50	47.3	7.1	4.4
Corn starch	315	72.8	54	72.8	54
Dicalcium phosphate	13	13	13	13	13
Soybean oil	25	25	25	25	25
Vitamin Mix V10001	10	10	10	10	10
L-Cystine	3	3	3	3	3
Potassium citrate	16.5	16.5	16.5	16.5	16.5
Mineral Mix 10026	10	10	10	10	10
Sucrose	350	172.8	172.8	172.8	172.8
Choline bitartrate	2	2	2	2	2
CP			21.5		21.5
WE				42.9	42.9
Total weight (g)	1055	858.1	858.1	858.1	858.1
Total energy (kcal)	4057	4057	4057	4057	4057

Note: The low-fat diet (LFD) was based on the Research Diets D12450B, and the high-fat diet (HFD) was based on the Research Diets D12451 [33]. The energy content of the LFD is estimated to be 3.85 kcal per gram, and the HFD is 4.73 kcal per gram. *Crataegus pinnatifida* (CP) and *Wolfiporia extensa* (WE) were added as powders. The composition and energy of CP and WE were calculated according to the Chinese Food Composition list [34].

## Data Availability

The original contributions presented in the study are included in the article, further inquiries can be directed to the corresponding authors.

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
