# Peer review of "The Effects of Crataegus pinnatifida and Wolfiporia extensa Combination on Diet-Induced Obesity and Gut Microbiota"

_foods, 2024, doi:10.3390/foods13111633_

Round 1
Reviewer 1 Report
Comments and Suggestions for Authors
This experimental paper explores an interesting and important question: the relationship between model animal consumption of Crataegus pinnatifida and Wolfiporia extensa and their physiological state, as well as the bacterial community in their gut. The study was well planned and carried out at a high scientific level. Many experimental parameters correspond to the best modern physiological publications.
The main shortcomings of this study are poor-quality statistical processing of data (the authors did not use methods of multiple comparison of samples) and errors in the description of the studied plant and fungal samples.
Disadvantages of the article.
1. It is better to formulate the title of the article in the form “The influence of something on something”, and not in the form of the result of this influence.
2. In the title of the article, the word “Bunge” means the name of the author who described this type of plant. This word should be removed from the title. Also, the abbreviation of this plant throughout the entire article should be not “Crataegus pinnatifida Bunge (CB)”, but “Crataegus pinnatifida (CP)” (line 20, 58, as well as throughout the article, including in the figures and tables). https://powo.science.kew.org/taxon/urn:lsid:ipni.org:names:724002-1
3. If the authors studied Poria cocos F.A. Wolf, then this is a synonym for the species Wolfiporia extensa (Peck) Ginns (1984). The authors must bring order to the taxonomy of the plant and fungus being studied: https://en.wikipedia.org/wiki/Wolfiporia_extensa. The abbreviated name of the fungus will not be “Poria cocos (PC)” (line 20, 60, and throughout the article, including in the figures and tables), but “Wolfiporia extensa (WE)”.
4. The introduction does not sufficiently characterize the plant and fungus. Readers should gain a detailed understanding of their chemical composition. A brief description does not allow us to understand the characteristics of the species being studied. I recommend adding a table to the Materials and Methods, in which you indicate 10–15 main components (according to literature data) of these species (table column names: structural formula of the substance and its name, content of the substance in the dry sample (%), physiological activity (in detail ), literature sources).
5. Table 1 does not contain a description of the composition of the diet for the “chow diet” group (line 89).
6. A space is required everywhere between the unit of measurement and the number (for example, lines 105, 109, 113).
7. Figure 1a should be made into a separate drawing and placed in Materials and Methods. Figure 1b needs to be enlarged 4 times in width and 2 times in height. It is necessary to ensure that readers see in which week of the experiment significant differences appear between the HFD group and other groups. Since it is necessary to apply multiple comparison methods (for example, Tukey's test), its results should be reflected not with asterisks, but with letters (a, ab, b, bс, c, cd, d) above the data points within one week of the experiment. The Tukey test should also be added to the description of statistical data processing methods.
8. Tukey's test should also be applied to compare all graphs (for example, Fig. 2b, 2d).
9. Readers don't have to memorize dozens of abbreviations. Authors must write the full and abbreviated names of all characteristics in the title of the figures (for example, line 197-199).
10. The authors compare only the first and second columns, as well as the second and third. Readers do not understand whether the first column is different from the third. This applies to all histograms in the article. The Tukey test or other multiple comparison method must be used.
11. Figures 4a, 4b, 4c, 4d show 6 animals per group, and the title of the figure indicates an 8-fold repetition.
12. In Figures 1-3, 5, 6, it is more correct to present all columns in the form of box analysis, as is done in Figure 4a, 4b, 4c, 4d.
13. In the titles of articles, you do not need to write all words with a capital letter (for example, lines 408, 420, 434).
Reviewer 2 Report
Comments and Suggestions for Authors
The manuscript is well organised, however, authors have to work the final version to minimise similarity impact to 20%. Introduction and discussion part need severe rephrasing and adding some recent references. The Materials and Methods needs much more extensive revision as similarity rate is even higher.
The second major issue is that the experimental design has 5 different groups. Authors, however, comment and provide only 3 groups in most of Tables/Figures. For this reason, major revisions are needed in parts such as results, discussion and conclusions.
Third point, is the diet analysis that needs further expalanations, energy content and chemical composition
Manuscript needs major revisions.
The subject of manuscript is obesity and the use of phytobiotics to moderate its effects on mice. Authors must explain the levels of diet phytobiotcs used and the duration of the trial in order to analyse their experimental design and justify their aim and the outcomes.
Round 2
Reviewer 1 Report
Comments and Suggestions for Authors
The authors did a good job. All flaws in the manuscript have been corrected.